# Relation-Aware Question Answering for Heterogeneous Knowledge Graphs

**Haowei Du[1,4], Quzhe Huang[1,4], Chen Li [5], Chen Zhang[1,4], Yang Li[5]**
**Dongyan Zhao[1,2,3,4*]**

[1]Wangxuan Institute of Computer Technology, Peking University
[2] The National Key Laboratory of Cross-Media General Artificial Intelligence
[3] Beijing Institute for General Artificial Intelligence (BIGAI), Beijing, China
[4] Institute for Artificial Intelligence, Peking University [5] Ant Group
duhaowei@stu.pku.edu.cn, {huangquzhe,zhangch,zhaodongyan}@pku.edu.cn
wenyou.lc@antgroup.com, ly200170@alibaba-inc.com

## Abstract

Multi-hop Knowledge Base Question Answering(KBQA) aims to find the answer entity in a knowledge graph (KG), which requires multiple steps of reasoning. Existing retrieval-based approaches solve this task by concentrating on the specific relation at different hops and predicting the intermediate entity within the reasoning path. During the reasoning process of these methods, the representation of relations are fixed but the initial relation representation may not be optimal. We claim they fail to utilize information from head-tail entities and the semantic connection between relations to enhance the current relation representation, which undermines the ability to capture information of relations in KGs. To address this issue, we construct a **dual relation graph** where each node denotes a relation in the original KG (**primal entity graph**) and edges are constructed between relations sharing same head or tail entities. Then we iteratively do primal entity graph reasoning, dual relation graph information propagation, and interaction between these two graphs. In this way, the interaction between entity and relation is enhanced, and we derive better entity and relation representations. Experiments on two public datasets, WebQSP and CWQ, show that our approach achieves a significant performance gain over the prior state-of-the-art. Our code is available on https://github.com/yanmenxue/RAH-KBQA.

## 1 Introduction

Knowledge Base Question Answering (KBQA) is a challenging task that aims to answer natural language questions with a knowledge graph. With the fast development of deep learning, researchers leverage end-to-end neural networks (Liang et al., 2017; Sun et al., 2018) to solve this task. Recently there has been more and more interest in solving complicated questions where multiple steps

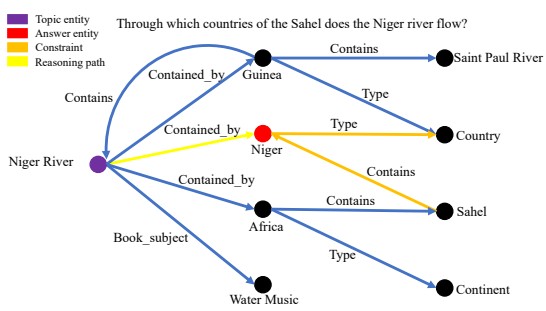

Figure 1: An example from WebQSP dataset. The yellow arrow denotes the reasoning path and the orange arrows denote two constraints "of the Sahel" and "country".

of reasoning are required to identify the answer entity (Atzeni et al., 2021; Mavromatis and Karypis, 2022). Such a complicated task is referred to as multi-hop KBQA.

There are two groups of methods in KBQA; information retrieval (IR) and semantic parsing (SP). SP-based methods need executable logical forms as supervision, which are hard to obtain in real scenes. IR-based approaches, which retrieves relevant entities and relations to establish the question specific subgraph and learn entity representation by reasoning over question-aware entity graph with GNN (Zhou et al., 2020), are widely used to solve multi-hop KBQA(Wang et al., 2020; He et al., 2021; Shi et al., 2021; Sen et al., 2021). These systems are trained without logical forms, which is a more realistic setting as these are hard to obtain. We focus on the retriever-reasoner approach to multi-hop KBQA without the addition of logical form data.

During the reasoning process of existing methods, the representation of relations are fixed but the initial relation representation may not be optimal. We claim the semantic connection between relations which is **co-relation** and the semantic information from head and tail (head-tail) entities to build relation representation is neglected.

We take one case in WebQSP dataset (Yih et al., 2015) as an example, where the question

---
*Corresponding Author

is "Through which countries of the Sahel does the Niger river flow?". In Figure 1, to answer the question, the model has to hop from the topic entity "Niger River" by the reasoning path "Contained_by". On the other hand, the candidate entity "Niger"(country) should connect to "Sahel" with relation "Contains" to satisfy the constraint "of the Sahel" to be the answer. So one crucial point for the model to retrieve the answer is to understand the meaning of relation "Contained_by" and "Contains". On one hand, the relation "Contained_by" and "Contains" have head-tail entities of type "location" like "Niger River", "Guinea", "Africa" and so on. The semantic information of entity type narrows down the meaning scope of relation "Contained_by" and "Contains" into locative relations. On the other hand, the massive intersection between head-tail entity sets of relation"Contained_by" and "Contains" implies the high semantic connection of these two relations, and we assume the representation learning for one of relations can enhance the other. If the model discovers the reverse relationship between "Contained_by" and "Contains", the understanding of one relation can help construct the representation of the other.

To utilize the semantic information from head-tail entities and the co-relation among relations, we propose our framework of unified reasoning of primal entity graph and dual relation graph. We construct the dual relation graph by taking all relations in the primal entity graph as nodes and linking the relations which share the same head or tail entities. In each hop, first we reason over primal entity graph by attention mechanism under question instructions, and then we do information propagation on dual relation graph by GAT (Veličković et al., 2017) to incorporate neighbor relation information into the current relation, at last we merge the semantic information of head-tail entities by the assumption of TransE (Bordes et al., 2013) to enhance the relation representation. We conduct the iteration for multiple rounds to ensure that the representation of relations and entities can stimulate each other. We do experiments on two benchmark datasets in the field of KBQA and our approach outperforms the existing methods by 0.8% Hits@1 score and 1.7% F1 score in WebQSP, 1.5% Hits@1 score and 5.3% F1 score in CWQ (Talmor and Berant, 2018), becoming the new state-of-the-art.

Our contributions can be summarized as three folds:

1. We are the first to incorporate semantic information of head-tail entities and co-relation among relations to enhance the relation representation for KBQA.

2. We create an iterative unified reasoning framework on dual relation graph and primal entity graph, where primal graph reasoning, dual graph information propagation, and dual-primal graph interaction proceed iteratively. This framework could fuse the representation of relations and entities in a deeper way.

3. Our approach outperforms existing methods in two benchmark datasets in this field, becoming the new state-of-the-art of retrieval-reasoning methods.

## 2 Related Work

Over the last decade, various methods have been developed for the KBQA task. Early works utilize machine-learned or hand-crafted modules like entity recognition and relation linking to find out the answer entity (Yih et al., 2015; Berant et al., 2013; Dong et al., 2015; Ferrucci et al., 2010). With the popularity of neural networks, recent researchers utilize end-to-end neural networks to solve this task. They can be categorized into two groups: semantic parsing based methods (SP-based) (Yih et al., 2015; Liang et al., 2017; Guo et al., 2018; Saha et al., 2019) and information retrieval based methods (IR-based) (Zhou et al., 2018; Zhao et al., 2019; Cohen et al., 2020; Qiu et al., 2020). SP-based methods convert natural language questions into logic forms by learning a parser and predicting the query graph step by step. However, the predicted graph is dependent on the prediction of last step and if at one step the model inserts the incorrect intermediate entity into the query graph, the prediction afterward will be unreasonable. SP-based methods need the ground-truth executable queries as supervision and the ground-truth executable query forms bring much more cost to annotate in practical. IR-based methods first retrieve relevant Knowledge Graph(KG) elements to construct the question-specific subgraph and reason over it to derive answer distribution by GNN. Zhang et al. (2018) introduce variance reduction in retrieving answers from the knowledge graph. Under the setting of supplemented corpus and incomplete knowledge graph, Sun et al. (2019) propose PullNet to

learn what to retrieve from a corpus. Different from EmbedKGQA (Saxena et al., 2020), which utilizes KG embeddings over interaction between the question and entities to reduce KG sparsity, we apply TransE in incorporating the head-tail entities information to enhance the relation representation. Shi et al. (2021) propose TransferNet to support both label relations and text relations in a unified framework, For purpose of enhancing the supervision of intermediate entity distribution, He et al. (2021) adopt the teacher-student network in multi-hop KBQA. However, it ignores the information from head-tail entities and semantic connection with neighbor relations to build the representation of current relation.

## 3 Preliminary

In this part, we introduce the concept of the knowledge graph and the definition of multi-hop knowledge base question answering task(KBQA).

### 3.1 KG

A knowledge graph contains a series of factual triples and each triple (fact) is composed of two entities and one relation. A knowledge graph can be denoted as $G = \{(e, r, e')|e, e' \in E, r \in R\}$, where $G$ denotes the knowledge graph, $E$ denotes the entity set and $R$ denotes the relation set. A fact $(e, r, e')$ means relation $r$ exists between the two entities $e$ and $e'$.

### 3.2 Multi-hop KBQA

Given a natural language question $q$ that is answerable using the knowledge graph, the task aims to find the answer entity. The reasoning path starts from *topic entity*(i.e., entity mentioned in the question) to the answer entity. Other than the topic entity and answer entity, the entities in the reasoning path are called *intermediate entity*. If two entities are connected by one relation, the transition from one entity to another is called one hop. In multi-hop KBQA, the answer entity is connected to the topic entity by several hops. For each question, a subgraph of the knowledge graph is constructed by reserving the entities that are within $n$ hops away from the topic entity to simplify the reasoning process, where $n$ denotes the maximum hops between the topic entity and the answer. To distinguish from the dual relation graph in our approach, we denote this question-aware subgraph as the primal entity graph.

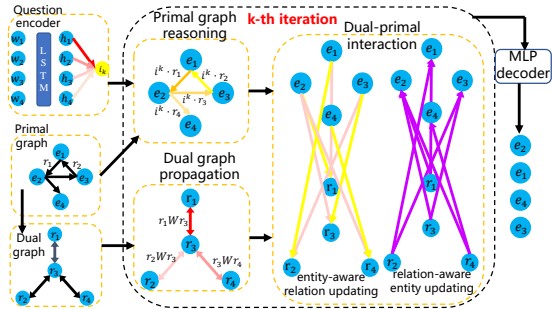

Figure 2: Method Overview. Our approach consists of four main components: question encoder, primal entity graph reasoning, dual relation graph propagation, and dual-primal graph interaction. The four components communicate mutually in an iterative manner.

## 4 Methodology

### 4.1 Overview

As illustrated in Figure 2 our approach consists of 4 modules: question instruction generation, primal entity graph reasoning, dual relation graph information propagation, and dual-primal graph interaction. We first generate $n$ instructions based on the question, and reason over the primal entity graph under the guidance of instructions, and then we do information propagation over the dual relation graph by GAT network, finally we update the representation of relations based on the representation of head-tail entities by TransE assumption,

### 4.2 Question Encoder

The objective of this component is to utilize the question text to construct a series of instructions $\{\mathbf{i_k}\}_{k=1}^n$, where $\mathbf{i_k}$ denotes the $k$-th instruction to guide the reasoning over the knowledge graph. We encode the question by SentenceBERT (Reimers and Gurevych, 2019) to obtain a contextual representation of each word in the question, where the representation of $j$-th word is denoted as $\mathbf{h_j}$. We utilize the first hidden state $\mathbf{h_0}$ corresponding to "[CLS]" token as the semantic representation of the question,i.e.,$\mathbf{q} = \mathbf{h_0}$. We utilize the attention mechanism to attend to different parts of the query and choose the specific relation at each step. Following He et al. (2021), we construct instructions as follows:

$$\mathbf{i^{(k)}} = \sum_{j=1}^{l} \alpha_j^{(k)} \mathbf{h_j} \tag{1}$$

$$\alpha_j^{(k)} = \mathbf{softmax}_j(\mathbf{W}_\alpha(\mathbf{q^{(k)}} \odot \mathbf{h_j})) \tag{2}$$

$$\mathbf{q^{(k)}} = \mathbf{W^{(k)}}[\mathbf{q}; \mathbf{i^{(k-1)}}] + \mathbf{b^{(k)}} \tag{3}$$

, where $l$ denotes the question length, $\mathbf{i}^{(0)}$ is initialized as $\mathbf{q}$, $\mathbf{W}_\alpha$, $\mathbf{W}^{(k)}$, $\mathbf{b}^{(k)}$ are learnable parameters.

### 4.3 Primal Entity Graph Reasoning

The objective of this component is to reason over entity graph under the guidance of the instruction obtained by the previous instructions generation component. First, for each fact $(e, r, e')$ in the subgraph, we learn a matching vector between the fact and the current instruction $i^{(k)}$:

$$\mathbf{m}^{(k)}_{(e,r,e')} = \sigma(\mathbf{i}^{(k)} \odot \mathbf{W_R r}) \tag{4}$$

, where $\mathbf{r}$ denotes the embedding of relation r and $\mathbf{W_R}$ are parameters to learn. Then for each entity $e$ in the subgraph, we multiply the activating probability of each neighbor entity $e'$ by the matching vector $\mathbf{m}^{(k)}_{(e,r,e')} \in \mathbb{R}^d$ and aggregate them as the representation of information from its neighborhood:

$$\hat{\mathbf{e}}^{(k)} = \sum_{(e,r,e') \in G} p^{(k-1)}_{e'} \mathbf{m}^{(k)}_{(e,r,e')} \tag{5}$$

The activating probability of entities $p^{(k-1)}_{e'}$ is derived from the distribution predicted by decoding module in the previous step. We concatenate the previous entity representation with its neighborhood representation and pass into a MLP network to update the entity representation:

$$\tilde{\mathbf{e}}^{(k)} = \mathbf{MLP}([\mathbf{e}^{(k-1)}; \hat{\mathbf{e}}^{(k)}]) \tag{6}$$

, where $\tilde{\mathbf{e}}^{(k)}$ denotes the representation of entity $e$ after incorporating neighbor entity information.

### 4.4 Dual Relation Graph Propagation

The objective of this part is to model and learn the connection between relations. To incorporate the semantic information from neighborhood relations to enhance the representation of current relation, we construct a dual graph for reasoning. The graph is constructed as follows: each node in dual graph denotes one relation in the primal graph. If two relations in primal graph share same head or tail entity, we believe they should have semantic communication and we connect them in the dual graph. We utilize GAT network to do information propa-

gation on dual relation graph:

$$\hat{\mathbf{r}}^{(k)} = \sum_{\bar{r} \in N(r)} \alpha_{r\bar{r}} \bar{\mathbf{r}}^{(k-1)} \tag{7}$$

$$\alpha_{r\bar{r}} = \frac{\exp(\mathbf{r W_{att} \bar{r}})}{\sum_{r' \in N(r)} \exp(\mathbf{r W_{att} r'})} \tag{8}$$

$$\tilde{\mathbf{r}}^{(k)} = \sigma(\mathbf{W_r}[\hat{\mathbf{r}}^{(k)}; \mathbf{r}^{(k-1)}] + \mathbf{b_r}) \tag{9}$$

where $\bar{\mathbf{r}}$ denotes the representation of neighbour relation $\bar{r}$, $\tilde{\mathbf{r}}^{(k)}$ denotes the representation of relation $r$ after integrating neighborhood relation information, $N(r)$ denotes the neighbor set of relation $r$ in dual relation graph, $\mathbf{W_{att}}$, $\mathbf{W_r}$ and $\mathbf{b_r}$ are parameters to learn.

### 4.5 Dual-Primal Graph Interaction

#### 4.5.1 Entity-Aware Relation Updating

The objective of this part is to incorporate the head-tail entity representation from the primal graph to enhance the representation of relations. Based on TransE method, we believe the subtraction between head-tail entity representation composes semantic information of the relation. So we use head entity representation $e_{tail}$ to minus tail entity representation $e_{head}$ as the representation $e_{fact}$. We encode the information of one relation from adjacent entities by passing all the fact representation $e_{fact}$ which contains it to an average pooling layer. Then we concatenate the neighbor entity information along with the previous representation and project into the new representation. The process above can be described as follows:

$$\mathbf{e}^{(k)}_{fact} = \mathbf{e}^{(k-1)}_{tail} - \mathbf{e}^{(k-1)}_{head} \tag{10}$$

$$\hat{\mathbf{r}}^{(k)} = \frac{\sum_{fact \in F_r} \mathbf{e}^{(k)}_{fact}}{n} \tag{11}$$

$$\mathbf{r}^{(k)} = \sigma(\mathbf{W_e}[\hat{\mathbf{r}}^{(k)}; \tilde{\mathbf{r}}^{(k)}] + \mathbf{b_e}) \tag{12}$$

where $F_r = \{f | f = (e, r, e') \in G, e \in E, e' \in E\}$, $n$ denotes the number of facts containing relation $r$, $\sigma$ denotes activating function, $W_e$ and $b_e$ are parameters to learn.

#### 4.5.2 Relation-Aware Entity Updating

In this part, we capture the information from adjacent relations to update entity representation by projecting the relation representation into entity space. Considering the semantic roles of head entity and tail entity in one relation are different, we adopt two separate projectors to handle the relations which hold the entity $e$ as head entity or tail

entity respectively:

$$\hat{\mathbf{e}}^{(\mathbf{k})} = \sum_{r \in R_e^{head}} \frac{\mathbf{W}_{\mathbf{head}}\mathbf{r}}{n^{head}} + \sum_{r' \in R_e^{tail}} \frac{\mathbf{W}_{\mathbf{tail}}\mathbf{r}'}{n^{tail}}$$

(13)

$$\mathbf{e}^{(\mathbf{k})} = \sigma(\mathbf{MLP}[\hat{\mathbf{e}}^{(\mathbf{k})}; \tilde{\mathbf{e}}^{(\mathbf{k})}])$$

(14)

where $R_e^{head} = \{r|(e, r, e') \in G\}$, $R_e^{tail} = \{r|(e', r, e) \in G\}$, $n^{head}$ and $n^{tail}$ denote the number of relations in $R_e^{head}$ and $R_e^{tail}$, $\mathbf{W}_{\mathbf{head}}$ and $\mathbf{W}_{\mathbf{tail}}$ are two separate groups of parameters to learn.

### 4.6 Decoding and Instruction Adaption

For simplicity we leverage MLP to decode the entity representation into distribution:

$$\mathbf{p}^{(\mathbf{k})} = \mathbf{softmax}(\mathbf{MLP}(\mathbf{E}^{(\mathbf{k})}))$$

(15)

where $\mathbf{p}^{(\mathbf{k})}$ denotes the activating probability of the $k$-th intermediate entity, and each column of $\mathbf{E}^{(\mathbf{k})}$ is the updated entity embedding $\mathbf{e}^{(\mathbf{k})} \in \mathbb{R}^d$. Considering the number of candidate entities is large and the positive sample is sparse among the candidates, we adopt the Focal loss function (Lin et al., 2017) to enhance the weight of incorrect prediction. Moreover we update the representations of instructions to ground the instructions to underlying KG semantics and guide the reasoning process in an iterative manner following Mavromatis and Karypis (2022). The representations of topic entities are incorporated by gate mechanism:

$$\mathbf{i}^{(\mathbf{k})} = (\mathbf{1} - \mathbf{g}^{(\mathbf{k})}) \odot \mathbf{i}^{(\mathbf{k})} + \mathbf{g}^{(\mathbf{k})} \odot \mathbf{h}^{(\mathbf{k})}$$

(16)

$$\mathbf{h}^{(\mathbf{k})} = \mathbf{W}_{\mathbf{i}}([\mathbf{i}^{(\mathbf{k})}; \mathbf{h}_{\mathbf{e}}^{(\mathbf{k})}; \mathbf{i}^{(\mathbf{k})} - \mathbf{h}_{\mathbf{e}}^{(\mathbf{k})}; \mathbf{i}^{(\mathbf{k})} \odot \mathbf{h}_{\mathbf{e}}^{(\mathbf{k})}]$$

(17)

$$\mathbf{h}_{\mathbf{e}}^{(\mathbf{k})} = \sum_{i \in \{e\}_q} \mathbf{e}_{\mathbf{i}}^{(\mathbf{k})}$$

(18)

where $\{e\}_q$ denotes the topic entities in the question, $W_i$ denote learnable parameters and $g^{(k)}$ dente the output gate vector computed by a standard GRU (Cho et al., 2014).

## 5 Experiments

### 5.1 Datasets

We evaluate our method on two widely used datasets, WebQuestionsSP and Complex WebQuestion. Table 1 shows the statistics about the two datasets.

| Datasets | train | dev | test | entities |
|---|---|---|---|---|
| WebQSP | 2,848 | 250 | 1,639 | 1,429.8 |
| CWQ | 27,639 | 3,519 | 3,531 | 1,305.8 |

Table 1: Statistics about the datasets. The column "entities" denotes the average number of entities in the subgraph for each question

| Models | WebQSP | | CWQ | |
|---|---|---|---|---|
| | Hits@1 | F1 | Hits@1 | F1 |
| GraftNet | 67.8 | 62.8 | 32.8 | - |
| PullNet | 68.1 | - | 45.9 | - |
| EmbedKGQA | 66.6 | - | - | - |
| TransferNet | 71.4 | - | 48.6 | - |
| Rigel | 73.3 | - | 48.7 | - |
| NSM | 74.3 | 67.4 | 48.8 | 44.0 |
| SQALER | 76.1 | - | - | - |
| REAREV | 76.4 | 70.9 | 52.9 | - |
| UniKGQA | 77.2 | 72.2 | 51.2 | 49.0 |
| Ours | **77.2** | **72.6** | **54.4** | **49.3** |

Table 2: Results on two benchmark datasets compared with several competitive methods proposed in recent years. The baseline results are from original papers. Our approach significantly outperforms NSM, where $p$-values of Hits@1 and F1 are 0.01 and 0.0006 on WebQSP, 0.002 and 0.0001 on CWQ.

**WebQuestionsSP(WebQSP)** (Yih et al., 2015) includes 4,327 natural language questions that are answerable based on Freebase knowledge graph (Bollacker et al., 2008), which contains millions of entities and triples. The answer entities of questions in WebQSP are either 1 hop or 2 hops away from the topic entity. Following Saxena et al. (2020), we prune the knowledge graph to contain the entities within 2 hops away from the mentioned entity. On average, there are 1,430 entities in each subgraph.

**Complex WebQuestions(CWQ)** (Talmor and Berant, 2018) is expanded from WebQSP by extending question entities and adding constraints to answers. It has 34,689 natural language questions which are up to 4 hops of reasoning over the graph. Following Sun et al. (2019), we retrieve a subgraph for each question using PageRank algorithm. There are 1,306 entities in each subgraph on average.

### 5.2 Implementation Details

We optimize all models with Adam optimizer, where the batch size is set to 8 and the learning rate is set to 7e-4. The reasoning steps are set to 3.

| Models | WebQSP | | CWQ | |
| --- | --- | --- | --- | --- |
| | Hits@1 | F1 | Hits@1 | F1 |
| P-Transfer | 74.6 | 76.5 | 58.1 | - |
| RnG-KBQA | - | 75.6 | - | - |
| DECAF* | 74.7 | 49.8 | 50.5 | 54.7 |
| DECAF(l) | 80.7 | 77.1 | 67.0 | 79.0 |
| Ours | 77.2 | 72.6 | 54.4 | 49.3 |

Table 3: Comparing with recent semantic parsing methods. DeCAF* and DeCAF(l) denote the DeCAF only with answer as supervision and DeCAF with T5-large as backbone.

The hidden size of GRU and GNN is set to 128.

## 5.3 Evaluation Metrics

For each question, we select a set of answer entities based on the distribution predicted. We utilize two evaluation metrics Hits@1 and F1 that are widely applied in the previous work (Sun et al., 2018, 2019; Wang et al., 2020; Shi et al., 2021; Saxena et al., 2020; Sen et al., 2021). Hits@1 measures the percent of the questions where the predicted answer entity is in the set of ground-truth answer entities. F1 is computed based on the set of predicted answers and the set of ground-truth answers. Hits@1 focuses on the top 1 entity and F1 focuses on the complete prediction set.

## 5.4 Baselines to Compare

As an IR-based method, we mainly compare with recent IR-based models as follows:

**GraftNet** (Sun et al., 2018) uses a variation of GCN to update the entity embedding and predict the answer.

**PullNet** (Sun et al., 2019) improves GraftNet by retrieving relevant documents and extracting more entities to expand the entity graph.

**EmbedKGQA** (Saxena et al., 2020) incorporates pre-trained knowledge embedding to predict answer entities based on the topic entity and the query question.

**NSM** (He et al., 2021) makes use of the teacher framework to provide the distribution of intermediate entities as supervision signs for the student network. However, NSM fails to utilize the semantic connection between relations and information from head-tail entities to enhance the construction of current relation representation.

**TransferNet** (Shi et al., 2021) unifies two forms of relations, label form and text form to reason over knowledge graph and predict the distribution of entities.

**Rigel** (Sen et al., 2021) utilizes differentiable knowledge graph to learn intersection of entity sets for handling multiple-entity questions.

**SQALER** (Atzeni et al., 2021) shows that multi-hop and more complex logical reasoning can be accomplished separately without losing expressive power.

**REAREV** (Mavromatis and Karypis, 2022) perform reasoning in an adaptive manner, where KG-aware information is used to iteratively update the initial instructions.

**UniKGQA** (Jiang et al., 2022) unifies the retrieval and reasoning in both model architecture and learning for the multi-hop KGQA.

We also compare with several competitive SP-based methods as follows:

**P-Transfer** (Cao et al., 2021) leverages the valuable program annotations on the rich-resourced KBs as external supervision signals to aid program induction for the low-resourced KBs that lack program annotations.

**RnG-KBQA** (Ye et al., 2021) first uses a contrastive ranker to rank a set of candidate logical forms obtained by searching over the knowledge graph. It then introduces a tailored generation model conditioned on the question and the top-ranked candidates to compose the final logical form

**DECAF** (Yu et al., 2022) Instead of relying on only either logical forms or direct answers, DECAF jointly decodes them together, and further combines the answers executed using logical forms and directly generated ones to obtain the final answers.

## 5.5 Results

The results of different approaches are presented in Table 2, by which we can observe the following conclusions: Our approach outperforms all the existing methods on two datasets, becoming the new state-of-the-art of IR-based methods. It is effective to introduce the dual relation graph and iteratively do primal graph reasoning, dual graph information propagation, and dual-primal graph interaction. Our approach outperforms the previous

| Quantile | 0-0.25 | 0.25-0.5 | 0.5-0.75 | 0.75-1 |
|---|---|---|---|---|
| NSM | 65.4 | 70.5 | 68.2 | 63.1 |
| Ours | 71.0 | 74.0 | 74.2 | 71.3 |

Table 4: Average F1 score on 4 groups of cases in WebQSP classified by relation number.

| Quantile | 0-0.25 | 0.25-0.5 | 0.5-0.75 | 0.75-1 |
|---|---|---|---|---|
| NSM | 65.6 | 66.7 | 69.3 | 65.6 |
| Ours | 69.7 | 72.6 | 75.5 | 72.8 |

Table 5: Average F1 score on 4 groups of cases in WebQSP classified by fact number.

| Quantile | 0-0.25 | 0.25-0.5 | 0.5-0.75 | 0.75-1 |
|---|---|---|---|---|
| NSM | 71.4 | 66.5 | 66.3 | 63.0 |
| Ours | 77.0 | 72.2 | 72.4 | 68.9 |

Table 6: Average F1 score on 4 groups of cases in WebQSP classified by the ratio between relation number and entity number.

| Quantile | 0-0.25 | 0.25-0.5 | 0.5-0.75 | 0.75-1 |
|---|---|---|---|---|
| NSM | 72.3 | 68.9 | 66.8 | 59.2 |
| Ours | 77.3 | 73.4 | 71.6 | 68.3 |

Table 7: Average F1 score on 4 groups of cases in WebQSP classified by the ratio between fact number and entity number.

state-of-the-art REAREV by 0.8% Hits@1 score and 1.7% F1 score in WebQSP as well as 1.5% Hits@1 score in CWQ. Compared to the Hits@1 metric, the F1 metric considers the prediction of the whole answer set instead of the answer that has the maximum probability. Our performance in F1 shows our approach can significantly improve the prediction of the whole answer set. Our significant improvement on the more complex dataset CWQ shows our ability to address complex relation information.

Comparing with SP-based methods in table 3, we can derive the following findings: 1. SP-based methods utilize ground-truth executable queries as the stronger supervision, which will significantly improves the KBQA performance with IR-based methods. 2. If losing the ground-truth executable queries supervision signal, the performance of SP-based methods will drop by a large degree. The SOTA SP-method DecAF, drops by 6 and 18 Hits@1 score on WebQSP and CWQ with answer as the only supervision. In the setting, our method outperforms DecAF by 3 and 4 Hits@1 score on WebQSP and CWQ. The ground-truth executable query forms bring much more cost and difficulty to annotate in practical. So the reasoning methods are important in real scenes where the ground-truth logical forms are hard to obtain. If the ground-truth executable queries are at hand, SP-based methods deserve to be take into consideration.

## 6 Analysis

In this section, we probe our performance on cases with different sizes and connectivity of dual relation graph and primal entity graph. Then we explore the benefit on relation and entity representation by introducing dual-primal graph interaction along with dual graph propagation. Furthermore, we do 3 ablation studies to evaluate different mod-

ules in our approach and explore our performance with different reasoning steps $n$.

### 6.1 Size and Connectivity of Dual and Primal Graph

In this part, we classify the cases of WebQSP dataset into 4 groups respectively based on the quantile of 4 metrics to ensure the same number of different groups: the number of relations, the number of facts, the ratio of relation number by entity number, the ratio of fact number by entity number. The number of relations and facts reflects the size and connectivity of dual relation graph, while the ratio of fact number by entity number reflects the connectivity of primal graph. The connectivity between primal entity graph and dual relation graph is embodied in the ratio of relation number by entity number.

From Table 3-6, overall, our approach derives better improvement over NSM on cases with larger size and high connectivity of dual relation graph and primal entity graph. With the larger size and higher connectivity, our dual graph propagation module incorporates more neighbor relation information to enhance the representation of current relation and the dual-primal interaction module exploits more abundant information from head-tail entities into relation representation.

### 6.2 Evaluation of Relation and Entity Representation

In this part, we explore the quality of the representation of relations and entities. By the assumption of TransE, the representation of relation should be translation operating on the embedding of its head entity and tail entity, which satisfies $e_{head} + r \approx e_{tail}$. So we choose the inner product of relation representation and head-tail entity embedding subtraction as the metric to evaluate the

quality of relation and entity representation:

$$Score_t = \frac{\sum_{n=1}^{N} \sum_{i=1}^{N_n} |\langle \mathbf{r_i^n}, (\mathbf{e_{i,tail}^n} - \mathbf{e_{i,head}^n}) \rangle|}{N \times \sum_{n=1}^{N} N_n \times d}$$

where $\langle \cdot \rangle$ denotes the inner product, $N_n$ denotes the number of facts in case $n$ and $d$ denotes the dimension of relation representation. The higher score means the relation representation is more relevant and consistent with the head-tail entities and the lower score denotes the relation representation space is more orthogonal to the entity space which is not ideal.

By Table 8, our approach significantly improves the TransE-score over NSM in cases with different sizes and connectivity of relation graph. It demonstrates our representations of relations and entities are much more corresponding to the semantic relationship between them. Incorporating head-tail entities information and the semantic co-relation between relations enhance the construction of relation representation, which stimulates the entity representation conversely. Moreover, the semantic consistency of our relation representation is better with a larger size and higher connectivity of relation graph. It illustrates the efficiency of our dual graph propagation and dual-primal graph interaction module for complex and multi-relational KGs.

### 6.3 Ablation Study

In this part, we do 3 ablation studies on WebQSP dataset to demonstrate the effectiveness of different components of our approach.

**Does dual relation graph information propagation matter?** In this ablation, we remove the dual relation graph propagation module, where the semantic connection of neighbor relation is not used to enhance the representation of current relation. By Table 9, we can see the performance drops by 0.4% score on Hits@1 and 1.5% score on F1. It shows our dual graph propagation module provides relevant semantic information from neighborhood relations which share the same head or tail entity to enhance the current relation.

**Does interaction between dual relation graph and primal entity graph matter?** In this ablation, we remove the interaction between dual graph and primal graph, where the semantic information of head-tail entities is not adopted to update the representation of relation connecting them. By table 9,

we can see the performance drops by 0.1% score on Hits@1 and 0.6% score on F1. It shows our interaction between primal and dual graphs incorporates semantic information of head-tail entities which is crucial to represent the corresponding relation.

**Can we use co-occurrence statistics of relations to replace the attention mechanism in dual graph propagation?** In this ablation, we employ co-occurrence of relations (Wu et al., 2019) to replace attention mechanism in Eq. 8. To be specific, let $H_i$ and $T_i$ denote the head-tail entity set of relation $r_i$ respectively, we compute the co-relation weights of $r_i$ and $r_j$ as follows:

$$\alpha_{ij} = \frac{(H_i \cup T_i) \cap (H_j \cup T_j)}{(H_i \cup T_i) \cup (H_j \cup T_j)} \qquad (19)$$

By table 9, we can see the performance drops by 0.9% score on F1. It shows introducing attention mechanism to weigh the co-relation between relations is more flexible and efficient than freezing the weights as the co-occurrence statistics. The weights of connection between relations should update as the reasoning process proceeds.

### 6.4 Different Reasoning Steps

For baselines which apply multiple hops reasoning, the reason step ranges from 2 to 4. We experiment on CWQ dataset with hops from 2 to 4 and freeze all other hyperparameters. Table 10 shows we gain stable improvement over the baseline with diverse reasoning steps. It demonstrates the robust efficiency of our iteratively primal graph reasoning, dual graph information propagation, and dual-primal graph interaction framework.

### 6.5 Case Study

We take an example from WebQSP dataset in figure 3 to show the efficiency of our dual graph information propagation module and dual-primal graph interaction module. The correct reasoning path contains 2 relations "Film_starring" and "Performance_actor" so one key point for the model to answer the question is to understand and locate these 2 relations. The baseline NSM fails to address the specific relation information and mistakes relation "Award_nomination" and "Award_nominee" for the meaning of acting, which predicts the wrong answer "Alan". However, considering that the type of head-tail entity for relation "Award_nomination" is film and award, the type of head-tail entity for relation "Award_nomination" is award and person,

| Quantile | Fact Number | | | | Relation Number | | | | |
|---|---|---|---|---|---|---|---|---|---|
| | 0-0.25 | 0.25-0.5 | 0.5-0.75 | 0.75-1 | 0-0.25 | 0.25-0.5 | 0.5-0.75 | 0.75-1 | Overall |
| NSM | 0.78 | 1.02 | 0.94 | 0.90 | 0.85 | 1.02 | 1.01 | 0.77 | 0.91 |
| Ours | 3.02 | 4.10 | 6.78 | 8.33 | 4.72 | 5.90 | 5.34 | 6.27 | 5.56 |

Table 8: Evaluating relation and entity representation for WebQSP dataset based on metric of TransE-score. We classify the cases into 4 groups by the number of relations and facts respectively to compute the average TransE-score and the "Overall" denotes the average score of the whole dataset.

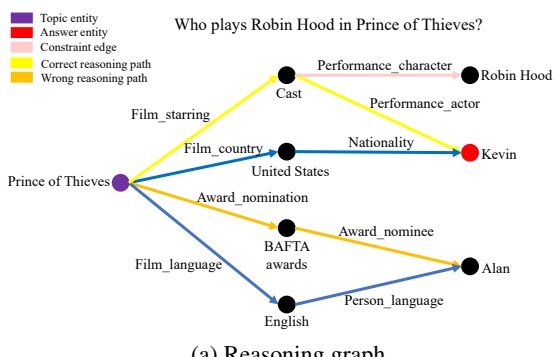

(a) Reasoning graph.

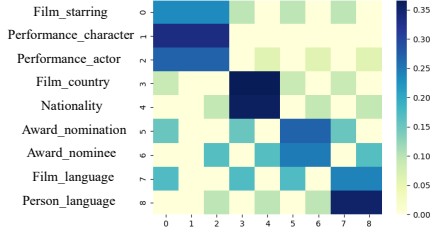

(b) Attention weights between relations in dual graph propagation.

Figure 3: An example from WebQSP dataset. The question is "Who plays Robin Hood in Prince of Thieves?". The yellow path denotes the reasoning path from topic entity "Prince of Thieves" to intermediate entity "Cast" then to answer "Kevin" and the pink edge denotes that "Cast" satisfies the constraint of character "Robin Hood".

| Model | Hits@1 | F1 |
|---|---|---|
| -Dual Graph Propagation | 76.8 | 71.1 |
| -Interaction | 77.1 | 72.0 |
| -Attention | 77.2 | 71.7 |
| Ours | **77.2** | **72.6** |

Table 9: Ablation study on WebQSP dataset.

| Model | Hits@1 | F1 |
|---|---|---|
| NSM | 48.8 | 44.0 |
| Ours(2) | 53.9 | 51.2 |
| Ours(3) | 54.4 | 49.3 |
| Ours(4) | 53.9 | 48.9 |

Table 10: Performance with different reasoning steps $n$ on CWQ dataset. The row "Ours(2)" denotes our model with $n = 2$.

their meanings can be inferred as nominating instead of acting by our dual-primal graph interaction module. Similarly, the information that entity types of head-tail entities for relation "Film_starring" are film and cast, entity types of head-tail entities for relation "Performance_actor" are cast and person, helps our model to address the meaning of these 2 relations as acting by dual-primal graph interaction. Moreover, our dual graph propagation module assigns important attention among these relations in the reasoning path like constraint relation "character", which incorporates neighborhood co-relation to enhance the learning of their representations.

## 7 Conclusion

In this paper, we propose a novel and efficient framework of iteratively primal entity graph reasoning, dual relation graph information propagation, and dual-primal graph interaction for multi-hop KBQA over heterogeneous graphs. Our approach incorporates information from head-tail entities and semantic connection between relations to derive better relation and entity representations. We conduct experiments on two benchmark datasets in this field and our approach outperforms all the existing methods.

## Limitations

We showed that our model is efficient in handling complex questions over KBs with heterogeneous graphs. However, we do not investigate the effect of pretrained knowledge graph embedding which contains more structural and semantic information from a large scale of knowledge graphs. In the future, we will probe the manner of incorporating pretrained KG embedding into our dual-primal graph unified reasoning framework.

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

# A  Example Appendix

This is a section in the appendix.