# OpenReview forum: "Relation-Aware Question Answering for Heterogeneous Knowledge Graphs"
_EMNLP/2023/Conference — EMNLP 2023 Findings_

### Official Review · Reviewer_qVC6 · 2023-08-03

**Typos Grammar Style And Presentation Improvements:** Section ‘A Example Appendix’ should b…
**Soundness:** 4

**Excitement:**

4: Strong: This paper deepens the understanding of some phenomenon or lowers the barriers to an existing research direction.

**Paper Topic And Main Contributions:**

This paper aimed to improve the Multi-hop KBQA performance, by incorporating the co-relation information. It proposed the concept of dual relation graph, in which node representing relations while edges representing entities, except for the primal entity graph. By interaction between the primal graph and the dual graph, this paper fuses the representation of both entities and relations. The main framework of this paper includes primal entity graph reasoning, dual graph information propagation, and dual-primal graph interaction. It evaluated the algorithm on two datasets, namely WebQuestionsSP and Complex WebQuestion, and show a new sota performance of KBQA.

**Questions For The Authors:**

The word 'Heterogeneous' is included in both the title and keyword list. However, not evident explanation about it is included. Besides, knowledge graph is naturally a heterogeneous graph. So is it necessary to emphasize it evidently in title and keyword list

**Reasons To Accept:**

1. It is interesting to consider the co-relation information that exist in many question patterns. The paper proposes the dual relation graph to include this kind of information into its framework, and obtain new sota.
2. Solid experiments are conducted, which helps explain why this framework works. It includes in detail the difference when the entity/relation graphs has different scales and connectivity, the influence of the dual graph introduction and some ablation study.

**Reasons To Reject:**

I saw no evident reasons to reject this paper.

**Reproducibility:**

4: Could mostly reproduce the results, but there may be some variation because of sample variance or minor variations in their interpretation of the protocol or method.

**Reviewer Confidence:**

4: Quite sure. I tried to check the important points carefully. It's unlikely, though conceivable, that I missed something that should affect my ratings.

---

> ### Author Rebuttal · Authors · 2023-08-29
>
> Thanks for appreciating our work. “heterogeneous” in the title denotes the nodes and relations in primal entity graph and dual relation graph are different. We will give its explanation in our paper.

---

### Official Review · Reviewer_h3Ex · 2023-08-11

**Typos Grammar Style And Presentation Improvements:** 1. Table 2's size should be modified …
**Soundness:** 3

**Excitement:**

3: Ambivalent: It has merits (e.g., it reports state-of-the-art results, the idea is nice), but there are key weaknesses (e.g., it describes incremental work), and it can significantly benefit from another round of revision. However, I won't object to accepting it if my co-reviewers champion it.

**Missing References:**

Zhang, Jing, et al. "Subgraph retrieval enhanced model for multi-hop knowledge base question answering." arXiv preprint arXiv:2202.13296 (2022).

Jiang, Jinhao, et al. "Unikgqa: Unified retrieval and reasoning for solving multi-hop question answering over knowledge graph." arXiv preprint arXiv:2212.00959 (2022).

**Paper Topic And Main Contributions:**

This paper aims to promote question-answering over knowledge graphs, such as Freebase. The motivation is to enhance relation understanding by fusing the semantic information of neighborhood relations and connected entities' types. The experiments conducted exhibit significant performance gains on two public datasets. The main contributions are fully utilizing information from entities and relations and the deep fusion between them.

**Questions For The Authors:**

A. I am confused about the motivation for employing semantic information of neighborhood relations. As shown in Figure 3, it is not difficult for the existing reasoning models to differentiate between "Award_nomination --> Award_nominee" and "Film_starring --> Performance_actor" based on the question. Rather, the real challenge lies in recognizing the "Performance_character -- Robin Hood" constraint when facing many "Cast" CVT nodes. However, it appears that the proposed method cannot alleviate this issue.
B. It appears that removing any of the sub-modules only results in a slight decline in performance. Therefore, why does combining all of them lead to the ultimate performance improvement?
C. In essence, you just add two extra sub-module to model the entity type and relation semantic based on the NSM framework. We could potentially achieve good performance by simply adding entity type representation to the existing entity representation, and aggregating neighborhood relation representation with the existing relation representation.

**Reasons To Accept:**

1. The authors propose an iterative reasoning method that effectively fuses semantic information of relations and type information of entities.
2. They thoroughly analyze the method and demonstrate the importance of incorporating information from neighborhood relations and connected entities.


**Reasons To Reject:**

1. The novelty of their method is restricted as it resembles NSM in general, with the only addition of extra information from relations and entities, through a corresponding sub-module.
2. Insufficient demonstration of performance gain from the proposed method due to the absence of some important baselines.
3. The writing style deviates from standard conventions, with non-standard formatting of tables and formulas.
4. Due to the lack of detailed implementation details, I am concerned about the reproducibility of this paper.

**Reproducibility:**

3: Could reproduce the results with some difficulty. The settings of parameters are underspecified or subjectively determined; the training/evaluation data are not widely available.

**Reviewer Confidence:**

4: Quite sure. I tried to check the important points carefully. It's unlikely, though conceivable, that I missed something that should affect my ratings.

---

> ### Author Rebuttal · Authors · 2023-08-29
>
> R1 : The contributions between our method and NSM are totally different. The novelty of NSM lies in the teacher-student framework where a teacher model is trained beforehand and provides the soft supervision for student model. This doubles the training time and we do not apply this framework for efficiency. We focus on the graph reasoning part which is essential in reasoning methods. We propose to incorporate semantic information of head-tail entities and co-relation among relations to enhance the relation representation for KBQA. To this end, we design the Dual Relation Graph Propagation module as well as Dual-Primal Graph Interaction module, and unify in an iterative manner. Compared with NSM, our method reduces the computation cost and outperforms by 2.9 Hits@1 and 5.2 F1 in WebQSP, as well as 5.6 Hits@1 and 5.3 F1 in CWQ, which demonstrates our efficiency.
>
> R2: SR [1] designs a plug-and-play subgraph retriever and utilizes NSM as graph reasoning. We focus on the improvement on graph reasoning instead of graph retriever, so we did not compare with SR. UniKGQA [2] unifies the retrieval and reasoning in both model architecture and learning for the multi-hop KGQA. Our method outperforms UniKGQA by 0.4 F1 on WebQSP, 3.2 Hits@1 and 0.3 F1 on CWQ. We will add this baseline in revised version.
>
> R3: Thanks for the suggestion. We will revise the table size and formula format in further version.
>
> R4: We conduct the experiments with 5 different random seeds and present the mean value. The p-value is less than 0.01 and our method significantly outperforms the existing SOTA of KBQA reasoning methods. We use SentenceBERT to encode the questions and relations.
> The number of hidden dimensions d is tuned among {64; 128}. We optimize the model with Adam optimizer, where the learning rate is set tuned among {5e-4; 7e-4; 9e-4} and the batch size is tuned amongst {8; 16; 32}. We tune the number of epochs amongst {30; 50; 70} and apply dropout regularization with probability tuned amongst {0.1; 0.2; 0.3}. We will add the details in Appendix of revised version.
>
> Q1: In Figure 3, it is crucial to understand the relation “Performance_character” to satisfy the constraint. The dual graph propagation module enhances the relation representation by the relevant neighborhood relations like “Performance_actor”. Moreover, the knowledge of entity “Robin Hood” which is a film character promote the relation learning.
>
> Q2: The ultimate improvement is not the simple accumulation of different modules. Considering different modules proceed iteratively, one module could bring some interference to other modules. However from Table 8 we can see different modules have irreplaceable effects in our method.
>
> Q3: The information (knowledge) of entities does not only contains entity type, but also identity, location or so, which are crucial to infer the relation. In Figure 3, the entity type “person” can not infer the precise meaning of relation “Performance_actor” considering many other relations connecting a film and person like “producer”. The knowledge that “Robin” is a film actor helps the model learn the relation. Moreover, Not only the head-tail entities but also co-relation from relevant relations enhance the representation of current relation by our dual graph propagation module.
>
> [1] Zhang, Jing, et al. "Subgraph retrieval enhanced model for multi-hop knowledge base question answering." arXiv preprint arXiv:2202.13296 (2022).
>
> [2] Jiang, Jinhao, et al. "Unikgqa: Unified retrieval and reasoning for solving multi-hop question answering over knowledge graph." arXiv preprint arXiv:2212.00959 (2022).

---

### Official Review · Reviewer_ngck · 2023-08-11

**Soundness:** 3

**Excitement:**

3: Ambivalent: It has merits (e.g., it reports state-of-the-art results, the idea is nice), but there are key weaknesses (e.g., it describes incremental work), and it can significantly benefit from another round of revision. However, I won't object to accepting it if my co-reviewers champion it.

**Missing References:**

Dedicated multi-hop QA benchmarks should be cited:
@inproceedings{yang2018hotpotqa,
  title={{HotpotQA}: A Dataset for Diverse, Explainable Multi-hop Question Answering},
  author={Yang, Zhilin and Qi, Peng and Zhang, Saizheng and Bengio, Yoshua and Cohen, William W. and Salakhutdinov, Ruslan and Manning, Christopher D.},
  booktitle={Conference on Empirical Methods in Natural Language Processing ({EMNLP})},
  year={2018}
}

@inbook{inbook,
author = {Dubey, Mohnish and Banerjee, Debayan and Abdelkawi, Abdelrahman and Lehmann, Jens},
year = {2019},
month = {10},
pages = {69-78},
title = {LC-QuAD 2.0: A Large Dataset for Complex Question Answering over Wikidata and DBpedia},
isbn = {978-3-030-30795-0},
doi = {10.1007/978-3-030-30796-7_5}
}


Actual SOTA that they don't beat is not cited:
@article{Yu2022DecAFJD,
  title={DecAF: Joint Decoding of Answers and Logical Forms for Question Answering over Knowledge Bases},
  author={Donghan Yu and Shenmin Zhang and Patrick Ng and Henghui Zhu and Alexander Hanbo Li and J. Wang and Yiqun Hu and William Wang and Zhiguo Wang and Bing Xiang},
  journal={ArXiv},
  year={2022},
  volume={abs/2210.00063},
  url={https://api.semanticscholar.org/CorpusID:252683172}
}

Alternative approaches to this task (namely retrieval for semantic parsing) are not cited:
@article{Ye2021RNGKBQAGA,
  title={RNG-KBQA: Generation Augmented Iterative Ranking for Knowledge Base Question Answering},
  author={Xi Ye and Semih Yavuz and Kazuma Hashimoto and Yingbo Zhou and Caiming Xiong},
  journal={ArXiv},
  year={2021},
  volume={abs/2109.08678},
  url={https://api.semanticscholar.org/CorpusID:237562927}
}

**Paper Topic And Main Contributions:**

Paper is about KGQA with a graph-based reasoning model. The authors claim to focus on multi-hop KBQA and argue for this setting to assess the claim that prior retrieval techniques do not leverage multi-node path information ("head-tail entities") nor information about each predicate neighborhood ("semantic connection between relations"). Contributions are:

1) an introduction to the type of graph structure information that presumably would benefit symbolic reasoning for KBQA. This includes an explanation of the graph processing necessary to to derive a "dual relation graph where each node denotes a relation in the original KG (primal entity graph) and edges are constructed between relations sharing same head or tail entities

2) a 4 module "graph reasoning" architecture that uses an LSTM question encoder and pipeline of bespoke networks (MLPs and GAT) to iteratively reasons over original graph, then derived graph, then their interaction.

3) an empirical comparison of their proposed approach with a handful of prior retrieval/graph reasoning approaches.

**Questions For The Authors:**

1) Can you give some motivation for the study of graph reasoning in the context of stronger alternatives? In other words, can you give more justification for the importance of the continued study of this approach?

2) Can you give more context on the selection of benchmarks? Given the paper's emphasis on multi-hop QA and on the specific benefits of the proposed approach in this setting, is there a reason that hard multi-hop benchmarks like HotPotQA, LC-QUAD2.0, and others were not considered?

**Reasons To Accept:**

There are a few strengths of this paper; in general, it is empirically sound and well written.
1) the authors do well to illustrate the challenge of multi-hop KBQA using both an illustrative figure and example question from WebQSP. Specifically, they make clear that systems have to levereage connectivity (hops) between relations to answer questions with constraints. This helps the reader understand why deriving some additional strucutural information about the graph might help

2) the authors give a detailed technical explanation of their proposed graph derivation and graph reasoning pipeline. Their proposed system is bespoke and technically complex, so this detail is required to understand what they propose.

3) the authors conduct a clear evaluation on two popular benchmarks (WebQSP and CWQ) and perform a useful set of ablations to reveal the benefits of alternative formulations of their system. Additionally, they present the results of these ablations in a very clear and easy to follow manner.

**Reasons To Reject:**

There are two main weaknesses of this paper:

1) the authors make a number of claims that are misleading or not supported by their results. For example,
1a) They claim state of the art performance on both CWQ and WebQSP but are 5-6% worse than the actual state of the art on both (e.g. Sept 2022's DecAF https://www.semanticscholar.org/reader/0751b6de6c07c52a324450e32ae6581d403603da).
1b) They claim to focus on multi-hop QA but study WebQSP and CWQ, two benchmarks that are widely known to have a majority of 1 and 2 hop questions. They do not evaluate on any KBQA datasets with explicit multi-hop subsets (e.g. HotpotQA, LC-QUAD 2.0, https://paperswithcode.com/dataset/lc-quad-2-0 and https://aclanthology.org/D18-1259/).
1c) The authors muddle the distinction between retrieval and graph reasoning approaches; the former is a broad category of very competitive approaches, while the latter is a fairly niche approach that has struggled to perform as well. Curiously, they do not evaluate against alternative retrieval based techniques that are explicitly designed for multi hop KBQA without using graph reasoning, such as HopRetriever (https://arxiv.org/abs/2012.15534), MultiHopDPR (https://openreview.net/pdf?id=EMHoBG0avc1), AISO (https://paperswithcode.com/paper/adaptive-information-seeking-for-open-domain). This is problematic because the field has largely moved away from the authors proposed bespoke graph-reasoning techniques towards direct retrieval and LM based pipelines that perform better. The authors don't address or argue for their chosen class of technique and instead simply omit this important context. It is fine to study an unpopular technique and it is clear that the authors conducted empirically rigorous science. It is a shame that the authors frame the work around a false claim of SOTA and on multi-hop QA, rather than present the work as an improvement in graph reasoning and argue for the continued study of that discipline. I can't give this paper the soundness score the empirical work deserves because the misleading claims the writing makes.

2) The ablations that the authors present seem to indicate that the benefits of the novel parts of their approach are fairly minimal. For example, removing dual-graph propagation only reduces performance by .5% Hit@1 to 1.5 F1; removing other bespoke formulations reduces performance by a similarly small margin (e.g. 0.1% 0.6% and 0.9%). They do not report error bars for their approaches, so it is not possible for us to tell whether this difference is statistically significant. This would be less of a concern if they were using a less complex architecture, in which the typical variance can be inferred from other works. The authors do mitigate this somewhat by providing a case study that additionally supports their claims. Ultimately, the claims that they make about the superiority of dual graph propagation and interactions are weak but defensible.

**Reproducibility:**

3: Could reproduce the results with some difficulty. The settings of parameters are underspecified or subjectively determined; the training/evaluation data are not widely available.

**Reviewer Confidence:**

3: Pretty sure, but there's a chance I missed something. Although I have a good feel for this area in general, I did not carefully check the paper's details, e.g., the math, experimental design, or novelty.

---

> ### Author Rebuttal · Authors · 2023-08-29
>
> R1: Regarding the comparison with other models:
>
> There are two mainstream approaches to solve KGQA: (i) parsing the question to executable KG queries like SPARQL, and (ii) grounding question and KG representations to a common space for reasoning. The former class needs
> ground-truth executable queries as stronger supervision than the sole answer entity, which is costly to obtain. So it is improper to compare the reasoning based methods using only answer as supervision with the parsing based methods using stronger supervision. Existing reasoning methods [1,2,4,5] do not compare with parsing methods for fair comparison. Our approach belongs to reasoning based methods and achieves the SOTA performance among the reasoning based methods which do not utilize ground-truth executable queries. The method DecAF [3] belongs to parsing methods using stronger supervision, so we did not compare with it. We will modify the expression and emphasize that we achieve the SOTA of reasoning based methods in KBQA.
>
> Regarding the selection of benchmarks:
>
> WebQSP and CWQ are two widely used benchmarks in multi-hop KBQA field. WebQSP requires up to 2-hop reasoning from knowledge base but CWQ contains more complex multi-hop questions and the questions require up to 4-hops of reasoning over the KG [1,2]. Following existing competitive reasoning methods [1,2,4,5], we choose WebQSP and CWQ as our experimental datasets. As for the two datasets mentioned in your review, HotpotQA is a dataset for multi-hop reading comprehension (MRC) which grounds the answer in several passages instead of the KG. It does not apply to KBQA. The original KB dump used to create the dataset LC-QUAD 2.0 is no longer available online [6] and recent methods do not utilize this dataset in experiments.
>
> Regarding retrieval in KBQA:
>
> Retrieval in KBQA is a stage before the graph reasoning, where a question specific subgraph is constructed from the full KG. We following [1,2] utilize PageRank-Nibble algorithm to extract the subgraph. Retrieval stage is independent from the reasoning stage and is not where most reasoning methods contributed in.
> Our contribution locates in the reasoning stage so we do not compare with other retrieval methods. HopRetriever, MultiHopDPR and AISO are methods for MRC which do not tackle KBQA task. LM based pipelines utilize the ground-truth executable queries as stronger supervision and the ground-truth executable queries are costly to get.
>
> R2: For each ablation we do 5 independent experiments with different random seeds and present the mean value. The variation scores of different seeds are within 0.1%-0.2% and the p-value is less than 0.01. The effectiveness of different modules in our approach are statistically significant. We will add the results related to significance test to our paper.
>
> Q1: The parsing based methods need the ground-truth executable queries as supervision and the performance will drop by a large degree with only answer as supervision. In Table 4 of DecAF [3], the performance drops by 6 and 18 Hits@1 score on WebQSP and CWQ. With answer as the only supervision, our method outperforms DecAF by 3 and 4 Hits@1 score on WebQSP and CWQ. The ground-truth executable query forms bring much more cost and difficulty to annotate in practical. So the reasoning methods are important in real scenes where the ground-truth logical forms are hard to obtain.
>
> Q2: WebQSP and CWQ are two widely used benchmarks in multi-hop KBQA field by both reasoning methods and parsing methods. WebQSP requires up to 2-hop reasoning from knowledge base, and CWQ contains more complex multi-hop questions and the questions require up to 4-hops of reasoning over the KG [1,2]. We following the recent methods [1,2,4,5] to choose the two benchmarks for fair comparison. As for the two datasets mentioned in your review, HotpotQA is a dataset in MRC which grounds the answer in several passages instead of the KG, and it does not apply to KBQA. The original KG dump used to create the dataset LC-QUAD 2.0 is no longer available online [6] and recent methods do not utilize this dataset in experiments. We will explore other KBQA benchmarks in revised version.
>
> [1] Improving Multi-hop Knowledge Base Question Answering by Learning Intermediate Supervision Signals. He et al. WSDM2021.
>
> [2] ReaRev: Adaptive Reasoning for Question Answering over Knowledge Graphs. Mavromatis et al. EMNLP2021.
>
> [3] DecAF: Joint Decoding of Answers and Logical Forms for Question Answering over Knowledge Bases. Yu et al. ICLR2023.
>
> [4] TransferNet: An Effective and Transparent Framework for Multi-hop Question Answering over Relation Graph. Shi et al. EMNLP2021.
>
> [5] Sqaler: Scaling question answering by decoupling multi-hop and logical reasoning. Atzeni et al. NIPS2021.
>
> [6] Modern Baselines for SPARQL Semantic Parsing. Banerjee et al. SIGIR 2022.

---

### Meta-Review · Area_Chair_fy5W · 2023-09-16

**Recommendation:** 3

**Metareview:**

This paper proposes a method for multi-hop knowledge base question answering. The idea is to use information from related relations and entities to improve reasoning over the KG. The paper provides experimental evidence and analysis supporting their method. However, reviewers also pointed out some issues regarding important baselines & comparisons that were missing.

---

### Decision · Program_Chairs · 2023-10-07

**Decision:**

Accept-Findings

**Comment:**

This paper proposes a method for multi-hop knowledge base question answering. The idea is to use information from related relations and entities to improve reasoning over the KG. The paper provides experimental evidence and analysis supporting their method. However, reviewers also pointed out some issues regarding important baselines & comparisons that were missing.